# Influence of 3D Printing Conditions on Some Physical–Mechanical and Technological Properties of PCL Wood-Based Polymer Parts Manufactured by FDM

**DOI:** 10.3390/polym15102305

**Published:** 2023-05-14

**Authors:** Irina Beșliu-Băncescu, Ioan Tamașag, Laurențiu Slătineanu

**Affiliations:** 1Faculty of Mechanical Engineering, Automotive and Robotics, “Stefan cel Mare” University, 720229 Suceava, Romania; 2Faculty of Machine Manufacturing and Industrial Management, “Gheorghe Asachi” Technical University of Iasi, 700050 Iași, Romania; lslati@yahoo.com

**Keywords:** wood-based biopolymer, surface quality, tensile strength, machinability

## Abstract

The paper investigates the influence of some 3D printing conditions on some physical–mechanical and technological properties of polycaprolactone (PCL) wood-based biopolymer parts manufactured by FDM. Parts with 100% infill and the geometry according to ISO 527 Type 1B were printed on a semiprofessional desktop FDM printer. A full factorial design with three independent variables at three levels was considered. Some physical–mechanical properties (weight error, fracture temperature, ultimate tensile strength) and technological properties (top and lateral surface roughness, cutting machinability) were experimentally assessed. For the surface texture analysis, a white light interferometer was used. Regression equations for some of the investigated parameters were obtained and analysed. Higher printing speeds than those usually reported in the existing literature dealing with wood-based polymers’ 3D printing had been tested. Overall, the highest level chosen for the printing speed positively influenced the surface roughness and the ultimate tensile strength of the 3D-printed parts. The cutting machinability of the printed parts was investigated by means of cutting force criteria. The results showed that the PCL wood-based polymer analysed in this study had lower machinability than natural wood.

## 1. Introduction

The ability to quickly generate complex surfaces and structures at lower costs and significantly lower material losses in the case of traditional mechanical processing technologies recommend 3D printing technologies for many industrial applications. There are several types of 3D printing processes, such as selective laser sintering (SLS), stereolithography (SLA), multi-jet fusion (MJF), digital light processing (DLP), digital light processing (DLP), fused deposition modelling (FDM), etc. FDM, also known as MEX (Material Extrusion) [1], is one of the most commonly used 3D printing processes because of the wide range of materials that can be processed/manufactured. The FDM process input parameters, such as the layer thickness, wall shell thickness, printing temperature, infill structure, infill density percentage, and printing speeds, strongly influence the mechanical proprieties of the printed products.

FDM is an emerging technology implemented in sectors such as the automotive, aerospace, medical, architecture, fashion, and food industries [2]. The main drawbacks reported for these technologies are the anisotropic nature and poor mechanical properties of the 3D-printed parts [2]. The principle of this manufacturing technique is that the wire material is heated and deposited layer by layer into the desired part shape. The part material must be pre-processed by hot melt extrusion to be transformed into filaments.

The literature provides multiple studies that analyse the influence of printing conditions and parameters and post-processing methods on the mechanical properties of 3D-printed parts, especially FDM [3,4,5,6,7,8,9,10,11]. Several researchers have contributed with comprehensive reviews on these issues [12,13,14]. Furthermore, the subject continuously develops due to the increasing interest in different industries, requiring more attention from the scientific community.

Voids usually appear between the deposited filament layers in the FDM printing process. These voids are believed to be one of the main causes of low tensile strength and anisotropy [15] and may also affect the 3D-printed parts’ cutting machinability. In the case of WPC (wood-based composite polymer) 3D-printed parts, it had been considered that wood fibres might encourage void formation. Comparing unfilled printed specimens with reinforced ones with natural fibres has shown a negative influence of the fibres on strength, while stiffness either increases slightly or remains constant [16].

The use of wood is increasing due to the growth of the world population, the development of new wood products, and the identification of new applications in various fields. Wood is a renewable and carbon-storing resource [16] with excellent properties but is limited to forest land. In recent decades, the wood demand increased significantly and overcame disposable supplies. Sustainability targets and growing environmental concerns have increased the demand for renewable and recyclable materials with compatible proprieties and behaviour/performance. In recent years, wood-based composite polymers (WPCs) have been gaining popularity [15,17,18,19]. These materials are composed of one or more natural wood chips, fibres, or flours and one or a mixture of polymers, most commonly thermoplastic polymers such as polyethene (PE), polylactide (PLA), or polypropylene (PP). Compared to natural timber products, WPCs present higher resistance to weathering and biological deterioration, thermal resistance, and expose sufficient strength for structural applications [20]. WPCs are mainly used for outdoor and indoor furniture, window and door frames, moulding, different construction purposes, and the automotive and marine industries [15,18,21]. The main drawbacks are the slightly higher prices and lower thermal resistance compared to natural wood.

The mechanical performance of WPCs is the main objective addressed by research in this field. Most of the research dealing with wood-based polymers analyses some mechanical proprieties for commercially available filaments [22,23] or develops and tests new wood composite filaments by mixing different amounts and types of wood fibres, polymers, additives, and fillers [15,16,24,25,26,27,28,29].

The most popular wood-based polymer type is obtained with a polylactic acid (PLA) polymer matrix and different percentages of wood fibres, dust, or chips. The performance of WPC material can be enhanced by using a proper combination of polymers and providing different fillers and additives. The research carried out in this field showed that beech sawdust can contribute to the reinforcement of flexural stress and tensile strength and that sawdust also helps reduce WPC density [21]. Additionally, WPCs are often brittle. Styrene and butadiene rubber (SBR), ethylene-propylene monomers leather (EPDM), or plastic elastomers can be used for toughening purposes [18]. Because of their strong flammability level, flame retardants, usually polyphosphate (APP), must be provided in their composition [19]. The addition of lignocellulosic fibres to WPC filaments was reported to lower the mechanical proprieties of the 3D-printed composites [27].

Hydrothermal degradation tests were performed [27] to establish its effect on the mechanical properties. The results showed that adding natural fillers and different levels of infilling resulted in a similar level of reduction in the properties. Additionally, the addition of natural fillers resulted in a slightly lower drop than the lowered infilling rate for tensile strength [27].

Results from another study [30] indicated that thickness swell, water uptake, mechanical strength, and stiffness increased, and elongation at break and impact energy decreased with an increasing wood fibre proportion.

The influence of the shape of wood particles on the mechanical proprieties of WPCs was also investigated. Huang et al. [29] showed that the shape and surface roughness of the wood particles, rather than the wood species, play an essential role in determining the properties of 3D-printed WPC products. Additionally, it was reported that wood particles with more rounded shapes and smoother surfaces are more suitable for obtaining a denser and stronger 3D-printed WPC product [29].

The machinability of wood–plastic composites has been approached by a relatively small number of studies. Most of these studies were carried on parts generated by other machining processes than 3D printing. Zhu et al. [31] explored the cutting performance of wood–plastic composites based on cutting forces, cutting temperature, surface quality, chip formation, and tool wear during peripheral milling experiments using cemented carbide cutters. The wood–plastic composites tested were processed by extrusion, moulding, and injection moulding. WPPC exhibited the highest cutting forces and cutting temperatures under the same cutting conditions, followed by WPEC and WPVCC. Wu et al. [32] had studied the helical milling performance of the WPC obtained by mixing poplar flour and polyethylene followed by extrusion at high temperatures. They reported that in WPC helical milling, the cutting force increases with increased spindle speed, cutting depth, and tool helical angle.

Biopolymers have attracted increased attention in recent years mainly because of their abundant and sustainable sources and versatile properties [2]. Biowood, produced by Rosa3D, is a wood-based composite biopolymer. The main components of Biowood filament are polycaprolactone (PCL), polyester, starch, lignin, natural resins, waxes and oils, natural fatty acids, cellulose, and natural fibres [33]. Polycaprolactone (PCL) is a biodegradable polyester with a low melting point of 60 °C [25] that is usually blended with other polymers. Studies [25] showed that natural fibres generally enhance polycaprolactone’s biodegradability and mechanical proprieties. Combining cellulose with polycaprolactone increased the tensile modulus but decreased the tensile strength of the composites [25]. Lignin is a natural polymer that binds cellulose fibres together, assuring stiffness for the wood-based polymer composites. Starch is not only used for binding and as a glue agent. The blending of starch with plastics has been reported to improve water resistance, processing properties, and mechanical properties [27].

Zgodavová K. et al. [23] have tested different thermoplastic materials for printing shield frames in terms of mechanical properties, geometric accuracy, weight, printing time, filament price, and environmental sustainability. Among them, they tested PHABiowood Rosa3D. The input parameters considered were the layer thickness, number of perimeters, extrusion width, infill density, and nozzle temperature. The tensile stress of the PHA Biowood varied from 10.8 MPa to 21.8 MPa, and the factors with significant influence over the mechanical properties were the infill and the interaction between the layer height and printing infill.

The aim of this paper was to highlight the results of some experimental studies dealing with the influence of some specific factors that characterise the 3D printing conditions of PCL wood-based polymer parts on some physical–mechanical and technological properties of the material incorporated in those parts. As input factors of the 3D-printing process, printing temperature, layer height, and printing speed were considered. Some physical–mechanical properties (weight error, fracture temperature, ultimate tensile strength) and technological properties (top and lateral surface roughness, cutting machinability) constituted output parameters that were subjected to the analysis. This study’s novelty consists in analysing the influence of some printing parameters (printing temperature, layer height, and printing speed) on some of the qualitative aspects and mechanical proprieties of the Biowood Rosa3D wood-based biopolymer parts generated by FDM. The values selected for the printing speed parameter were significantly superior to those usually tested in previous research in this field or those recommended by the filament producer. Another novelty aspect of this study is the slot milling machinability analysis by means of cutting force levels of the FDM-printed parts.

Even if complex shape parts can be generated by additive manufacturing, there are several situations where 3D-printed parts may require future processing.

The use of cutting as a secondary processing operation for 3D-printed parts can address for parts with high functional and tolerance requirements. Usually, the FDM parts achieved accuracies of ±0.5 mm for desktop printers and ±0.2 mm for industrial printers, and with CNC machining, accuracies of ±0.05 mm can be obtained.

Another reason for combining the two technologies is productivity. Even with the recent advancements in 3D-printing technology, printing speed is still a major drawback for considering these technologies for industrial applications. By considering cutting technologies for some of the part features, the machining time of the parts can be significantly improved.

This study can be a starting point for other researchers that aim to establish the proper printing conditions for PCL wood-based polymers and for industry agents interested in developing biodegradable wood-like products.

## 2. Experimental Setup

In Figure 1, a schematic representation of the experimental program used in the study is presented. The model offers information about the input parameters, equipment, procedures, and the investigated parameters considered in the study.

The parts, in the form of test specimens with dimensions according to ISO 527-2 1B, were manufactured using a BambuLab X1C 3D printer and then tested and analysed from four perspectives:Surface quality by obtaining values for surface roughness (Sq) using a Mahr CWM 100 profilometer;Tensile strength, obtaining values for UTS but also for the temperature at the time of specimen rupture;Analysis of the density variation of the resulting parts in terms of weight;Machinability of the parts, where the values for the components Fx, Fy, and Fz of the cutting force were obtained using a Kistler type 9257B dynamometer.

A full factorial experiment was considered to achieve the desired research objectives. The independent variable factors that were changed in the experimental procedure were the following: the printing temperature Tp (°C), the layer height, hl (mm), and printing speed, sp (mm/s). The values of the input factor levels selected in this study for each of them are presented in Table 1.

The results obtained under experimental conditions according to the DOE 3^3^ factorial design, were analysed to obtain variation plots, and then ANOVA was applied to determine the factors with statistically significant influence.

As output interest parameters of the proposed study, the following parameters were considered:-Weight error (%);-Arithmetical mean height, Sa (µm) of the top and lateral surfaces of the specimens;-Ultimate tensile strength UTS (MPa);-Fracture temperature T (°C);-Cutting force components Fx (N), Fy (N), and Fz (N).

The advancements in 3D printing equipment have opened new opportunities in terms of reducing the printing time. The producers of 3D printers have focused on addressing one of the main drawbacks of additive manufacturing technologies, which is the printing time. Printing time is directly proportional to the printing speed that can be achieved. Therefore, in recent years, new 3D printers with higher printing speed facilities were produced. Even if high printing speeds can be achieved by using these new 3D printers available in the market, the testing of these capabilities is still limited. In this study, significantly higher printing speeds than those usually reported in the scientific literature were considered.

The factor levels were chosen to preserve the randomness of the results. In the case of temperature, the minimum level was chosen to be 175 °C, the second level 190 °C, which is most often used in FDM 3D printing especially for biopolymers (such as PLA—polylactic acid), and 220 °C, 10 °C more than the manufacturer’s recommendation.

In terms of layer height, level 1 of 0.2 mm was chosen because it is the most common in the literature, 0.4 mm because the nozzle used has a diameter of 0.6 mm (dimensions suggested by the filament manufacturer), and the layer height represents under 75% of the nozzle diameter. The middle value of 0.28 mm was chosen to be able to observe inter-layer overlap and part density variation when the levels did not have a multiple character.

In terms of printing speed, high random speeds in the range 150–300 mm/s were chosen. These values were chosen because this range is less studied and the printer used, being core XY, allows printing at high speeds obtaining high qualities, comparable to 3D printing at low speeds.

The choice of level values was made in such a way that the midpoint was not close to the end values, thus avoiding the possibility of intercalation of mean effects.

### 2.1. Materials

Biowood is a raw polymer consisting of only renewable resources. The test samples used in the experiments were produced by Rosa3D Filaments (Poland). The main components of this wooden thermoplastic polymer filament are the following: polycaprolactone (PCL), polyester, starch, lignin, natural resins, waxes and oils, natural fatty acids, cellulose, and natural fibres [33]. The wood fibre content is considered to facilitate mechanical processing. In Table 2, the main physical properties of biowood polymer are presented. According to the filament producer, biowood filaments require low extrusion temperatures, between 170 and 210 °C. Moreover, printing speeds in the range 60–80 mm/s and build platform temperatures of 30–50 °C are recommended [33].

### 2.2. Sample Preparation and Equipment

Experimental tests were conducted considering standardised tensile test specimens ISO 527 Type 1B. The probes had the geometry and dimensions presented in Figure 2. For these studies, 100% infill specimens were considered.

Specimens were manufactured on an FDM Desktop enclosed printer type X1-Carbon Combo produced by Bambu Lab (Austin, TX, USA) (Figure 3). The printer has a lidar resolution of 7 μm, 20 m/s² acceleration, and a maximum speed of 500 m/s, and it works with a Prusa-type slicer. A hardened steel nozzle of 50 HRC with a diameter of 0.5 mm was used. The weight of the specimens was determined using an analytical balance produced by Kern (Balingen, Germany) type ADB 200-4 with a resolution of 0.0001 g. A 100% infill for all the tested samples was considered. The build platform temperature was set to 35 °C.

The theoretical part weight was determined by calculating the theoretical volume based on the nominal dimensions of the ISO 527-2 1B specimen and after multiplying it with the density provided by the producers of Rosa Biowood filaments in the technical data sheet. The estimated theoretical weight was used to determine the weight error for the 3D-printed parts. The weight error was calculated as the difference between the theoretical and the measured weight and divided by the theoretical weight as follows
(1)$$\varepsilon_{w}=\frac{\left(w_{t}-w\right)}{w_{t}}\;\cdot \;100\;\left[\%\right];$$

The tensile strength of the specimens was measured using experimental equipment (Figure 4) previously designed and executed within the Faculty of Mechanical Engineering, Automotive, and Robotics at the “Stefan cel Mare” University of Suceava, Romania. The tensile strength measuring device comprises specimen grips mounted on the crossheads of the tensile testing device body. The drive system controls the up or down motion of the moving crosshead. Sensors measure the specific elongation and traction force. After the amplifier amplifies the signal, the measuring results are introduced to a computer via a data acquisition device and processed by specialised software.

Fracture temperatures can be used to analyse the energy levels absorbed by the specimen material and the strain developed in the material before the rupture. The fracture temperature was measured using a high-speed thermal camera produced by Flir type X6540sc, produced by Teledyne FLIR (Wilsonville, OR, USA) (Figure 5), with an accuracy of ±1 °C/1%. The data provided by the camera were analysed and processed using Research IR specialized software. The maximum temperature before the rapture of the samples was retained and analysed in this study.

Surface quality was investigated by the surface area roughness parameter Sa (arithmetical mean height). According to ISO 25178, this parameter expresses the arithmetic mean of the height’s absolute value from the surface’s mean plane [35]. It is known that the most frequently used parameter for characterising the surface texture in a section through the machined surface of a part is the average arithmetic deviation Ra of the evaluated profile. In many situations, only values for the roughness parameter Ra are prescribed in part technical drawings. It is appreciated that, in relation to other roughness parameters, the Ra parameter provides the most information regarding the future operating behaviour of the surface it characterises. When the question arises of evaluating the roughness of a specific surface, the roughness parameter Sa has a similar meaning and importance to that of the roughness parameter Ra in the case of the profile of a surface in a certain section through the workpiece.

Because of the specific way the FDM printing processes are carried out, the printed parts’ top surface and lateral surfaces will expose different surface textures. These textures are a result of how the melted material layers are deposited. That is why both surfaces were considered. The measurements were carried out on three different surface areas, and the average value was determined and used in the study.

Sa surface roughness values were obtained using the Mahr CWM 100 confocal microscope and white light interferometer, produced by Mahr GmbH, Gottingen, Germany (Figure 6a), and surface topography (Figure 6b) was analysed using the related Mahrsurf MfM software Version 7.4.8676.

In Figure 7, the end milling setup for the cutting machinability testing is presented. The machining tests were carried out on a Diy CNC router. The cutting forces’ magnitude was measured using a Kistler dynamometer (produced by Kistler Group, Wien, Austria) type 9257B. The cutting parameters used for the machining tests were the following: cutting speed—150 m/min, cutting feed—800 mm/min, and depth of cut—ap = 1.5 mm. The cutting tool used was a two-flute end mill with a diameter of 3.17 mm made of ultrafine carbide Co10%, produced by Jiangsu Weixiang Tools Manufacturing Co., Ltd., Zhenjiang, China (Figure 7b). The obtained graphs for the cutting forces were processed and analysed in the related specialised software Dynoware version 3.3.1.0.

The experimental data obtained were analysed using a trial version of the DOE statistical software Minitab.

## 3. Results

The experimental results of the main output parameters investigated in this study are presented in Table 3.

### 3.1. Part Weight and Weight Error

Figure 8 presents the main effects and interaction plots obtained for the part weight.

The results show that the specimens printed with layer heights of 0.28 mm have a significantly higher weight and a smaller weight error than those with layer heights of 0.2 mm and 0.4 mm.

An explanation could be that this is caused by how the Prusa slicer determines the extrusion width and, more precisely, the overlapping between the extrusion lines when the height of the part is not an integer multiple of the layer height value. The overlap factor greatly impacts the FDM parts’ voids’ volume, conducting denser structures and lower weight error for the FDM-printed parts. Besides the layer height, the printing temperature is another important factor strongly influencing the part weight and weight error. This parameter influences the printed material’s thermal expansion, fluidity, layer adhesion, hardness, and tensile properties. Experimental results show a minimum weight error for the parts printed with a temperature of 220 °C which exceeds the range recommended by the Biowood filament producers. At lower printing temperatures, the material does reach the proper fluidity and causes bad adhesion and voids between the extrusion lines and layers.

Table 4 presents the analysis of variance (ANOVA) carried out to analyse how the selected input factors affect the experimental values obtained for the part weights. The test shows that with a 95% confidence interval, none of the inputs are statistically significant.

### 3.2. Sa Surface Roughness Parameter

#### 3.2.1. Roughness of the Top Surface of the Specimens

Figure 9 presents the influence exerted by the selected input parameters on Sa surface roughness of the top surface of the specimens. As it can be observed the printing temperature and layer height have a strong influence on the Sa surface roughness parameter variations.

Figure 10 presents the isometric images of the surface topography of the top surfaces of the 3D-printed parts obtained using the Mahr CWM 100 white light interferometer and confocal microscope. It can be seen that higher printing temperatures result in better layer adhesion and fewer pores. Additionally, when higher temperatures and higher layer heights are used, the upper top surfaces of the specimen expose significantly higher surface asperities that result from over-extrusion and signalise a bad material flow.

The ANOVA test carried out for the top surface Sa roughness parameter is presented in Table 5. The test result indicates that all the input parameters investigated are statistically significant, with a reliability coefficient of 0.95.

#### 3.2.2. Roughness of the Lateral Surface of the Specimens

Even if in the scientific literature [16] it is stipulated that increasing the printing speed is chosen at the expense of lower surface quality, in this study, for the PCL wood-based biopolymer investigated, the results show contrary aspects (Figure 9 and Figure 11). The printing speed exhibits a relatively low influence over the Sa surface roughness parameter measured for the top surface of the tested samples. Lower surface roughness values for the lateral surfaces of the samples were obtained for the parts printed with the highest level chosen for the printing speed sp = 300 mm/s. The arithmetical mean height roughness parameter increases with the increase in printing speed but tends to decrease after a certain value.

Figure 12 presents isometric images of the lateral surface texture of the specimens printed with a printing temperature of Tp = 190 °C and with a layer height of 0.4 mm at different printing speeds. The arithmetic means indicate that the height of the asperities is significantly lower when the highest level of the printing speed is adopted. 

These could be a result of the rapid cooling of the melted deposit layers due to the ventilation effect associated with the high velocity of the nozzle.

The ANOVA test carried out for the Sa roughness parameter measured for the lateral surfaces (Table 6) of the 3D-printed parts indicates that the layer height is statistically significant with a reliability coefficient of 0.95 (α = 0.05). This is also sustained by the Pareto graph (Figure 11b), which indicates that the interest parameter Sa variation is likely attributable to the layer height parameter variation.

### 3.3. Ultimate Tensile Strength

Figure 13 presents images of the fracture surfaces obtained in the tensile strength tests. The fracture appearance presents different proportions of brittle or ductile failure modes. It could be observed that the specimens obtained at higher printing temperatures exposed higher percentages of ductile fracture. This means that by using higher printing temperatures, the parts will have more toughness.

This was also reflected in the ultimate tensile strength values obtained in the tensile strength tests that were carried out. The printing temperature exposed a significant influence on ultimate tensile strength values. In the Pareto chart (Figure 14b), the level of significance of each input factor chosen for this study can be analysed. The results show that among the studied factors, the printing speed and the interaction between the printing temperature and layer height are statistically significant at the 0.05 level.

This result is also sustained by the analysis of variance carried out (Table 7). According to the ANOVA test, the printing speed is statistically significant for the ultimate tensile strength variation with a reliability coefficient of 0.95.

In this study, significantly higher printing speeds than those usually reported as being studied in the scientific literature (range 15–170 m/s) [1,3] were used. Higher printing speeds can prevent the alteration of biocomponents of the filaments due to intense exposure to high temperatures. Even if it is a general belief that higher printing speeds conduct weaker structures due to insufficient cooling time between layers and bad layer adhesion, the results obtained in this study indicate that a higher printing speed significantly increases the ultimate tensile strength and the fracture temperature of the printed parts (Figure 14 and Figure 15).

### 3.4. 3D-printed Parts Machinability

The cutting machinability of Biowood Rosa3D printed parts was also investigated. Slot milling tests were carried out, and the cutting force values obtained were compared with the ones achieved by machining in identical condition pinewood and beech wood samples. Pine and beech wood were selected as representatives for the soft and hard wood categories. A measurement of the cutting force components for slot milling operations of some samples from three distinct wooden materials was carried out, one of which was the biowood. The tests were carried out on a three-axis DIY milling router-type machine tool, using a two-flute tungsten carbide end mill type 10113117 produced by Weix tools, China. The geometry of the active zone of the end mill is typical for wood bits. As for cutting conditions, the following values were chosen: ap = 1.5 mm for depth of cut, f = 800 mm/min for cutting feed, and vc = 150 m/min for cutting speed.

By machining the samples obtained by FDM 3D printing of Biowood Rosa filaments, significantly higher cutting forces were obtained (Figure 16). The average cutting forces generated by machining Biowood Rosa samples were up to 10× higher than those obtained by end-milling softwood samples and up to 2.5× higher than those obtained for the hardwood samples.

Figure 17 presents the main influence of the 3D printing input parameters analysed in the study over the cutting force components. The printing temperature and layer height positively affect the cutting force components’ magnitude. Printing speed negatively influences the machinability of Biowood Rosa parts according to the force-cutting criteria. Even if the material becomes more ductile because of the exposure to high temperatures and therefore requires higher efforts to be machined, at high printing temperatures, over-extrusion phenomena could be observed by analysing the top surface topography of the specimens. This phenomenon can lead to weak structural bonds and is conducive to lower values for the cutting forces. 

### 3.5. Nonlinear Regression Analysis

Through the mathematical processing of the experimental results, it became possible to determine some empirical power, function-type mathematical models. With these empirical mathematical models, additional information was obtained regarding the order of influence and the intensity of the influence exerted by some factors on the output parameters of the investigated process. Microsoft Excel software was used for the mathematical processing of the experimental results. In this way, the following empirical mathematical models were obtained:-For the lateral surface roughness (standard error of the regression S = 6.0487, correlation coefficient R = 0.5039):
Sa = 7.3154 · Tp^0.4558^hl^0.2210^sp^−0.0747^ [µm];(2)

-For the ultimate tensile strength (standard error of the regression S = 2.076, correlation coefficient R = 0.6140):

UTS = 0.08885 T^0.7137^hl^0.1974^sp^0.2708^ [MPa];(3)

-For the part weight (standard error of the regression S = 0.0629, correlation coefficient R = 0.2167):

W = 9.5844 · T^0.06157^hl−^0.004079^sp^0.000083^ [g];(4)

Figure 18 shows normally distributed data for the regression equations determined for the lateral surface roughness of the printed parts and the ultimate tensile strength interest parameters. Additionally, the distances between the residuals versus their expected values for the regressions are relatively small. The parameter estimation errors are presented in Table 8, Table 9 and Table 10. The small values of the coefficient standard error (SE) indicate a precise estimation.

By examining the mathematical model corresponding to the lateral surface, Sa roughness parameter, it could be seen that these parameters will register an increase when the printing temperature TP and layer height hl increase and decrease with the increase in the printing speed sp. The printing temperature Tp exerts the most substantial influence on the Sa parameter, which, in the empirical mathematical model, corresponds to the highest value of the corresponding exponent compared to the values of the exponents attached to the rest of the analysed process input factors.

It should be noted that increasing the value of any of the three factors considered will increase the ultimate tensile strength UTS because the values of all exponents are positive. The printing temperature Tp is also the input factor with the strongest influence in the ultimate tensile strength UTS because, in this case, the value of the exponent attached to this factor also has the maximum value of the values of the exponents of the other input factors studied. An explanation of the increase in the value of the UTS parameter when increasing the printing temperature Tp could result from better adhesion of the deposited layers due to the higher values of the printing temperature.

The three input factors have a relatively small influence on the weight output parameter. This finding is based on the very low values of the exponents obtained for the input factors in the empirical mathematical model corresponding to the parameter W. However, it can be observed that in this case, the strongest influence also seems to be exerted by the printing temperature Tp, whose exponent has the maximum value.

## 4. Discussion

Even when 100% infill was set as the printing condition for the specimen manufacturing process, the resulting parts’ weight was smaller than the theoretical weight (determined by the theoretical volume and the material density value provided by the filament producers). The weight error calculated for the specimens ranged between 5.9–7.6%.

The surface roughness parameter Sa measured on the top surfaces of the 3D-printed samples ranged between 13.8 and 149.7 (μm). For this output parameter, all of the considered input factors were reported as statistically significant according to the ANOVA test carried out. Even if the printing temperature levels tested in this study did not exceed the temperature range recommended by the filament producer, the area surface roughness parameter Sa of the top surfaces of the samples printed at 190 °C recorded an average increase of 152%, and those printed at 220 °C showed an average increase of 347% compared with those printed with a temperature of 175 °C. The layer height parameter also exposed a similar influence on the Sa roughness of the top surfaces of the printed parts. The roughness parameter had an average increase of 176% when the layer height was set at 0.28 mm and an average increase of 303% when a layer height of 0.4 mm was adopted compared with that resulting from parts printed with a layer height of 0.2 mm. These variations result due to over-extrusion caused by inefficient flow rates.

The surface roughness parameter Sa values measured on the lateral surfaces of the printed specimens ranged between 35.34 and 57.92 (μm). According to the main effect plots (Figure 11a), the printing conditions that assured a better surface roughness were the lowest value of the printing temperature (175^°^), the layer height of 0.28 mm, and the maximum value of the printing speed (300 m/s). Among the input factors investigated, only the layer height tested as statistically significant according to the ANOVA test (Table 6).

The ultimate tensile strength values obtained were in the range of 7.5–19.06 MPa. Significant correlations were found between printing speed, mechanical strength (ultimate tensile strength), printing temperature–layer height interaction, and mechanical strength.

The machinability was investigated using cutting forces criteria. Machinability is rated relative to the results achieved for a representative/reference material. To evaluate the machinability of Biowood printed parts, machining tests were carried out in similar conditions for pinewood and beech wood as representatives of softwood and hardwood materials. The average cutting forces generated by machining Biowood Rosa samples were up to 10× higher than those obtained by end-milling softwood samples and up to 2.5× higher than those obtained for the hardwood samples. The printing temperature and layer height tend to positively affect the cutting force components’ magnitude, while printing speed negatively influences the machinability of Biowood Rosa printed parts.

## 5. Conclusions

Biopolymers are a natural alternative to synthetic polymers that exhibit reduced carbon dioxide emissions in their synthesis. In recent decades, more and more emphasis has been placed on using biopolymers for various medical, food, and industrial applications.

Few studies have been carried out on testing the capabilities of wood biopolymer composites. Most of these studies usually address only the mechanical properties of WPC. The present research explores the effect of printing temperature, layer height, and printing speed on surface quality, tensile performance, and cutting machinability of parts obtained by FDM printing of Rosa3D Biowood filament. Biowood produced by Rosa3D is a wood-based composite biopolymer obtained by amalgamating wood fibres in a polycaprolactone PCL and polyester polymeric matrix and adding fillers (starch, lignin) and additives (natural resins, waxes, and oils, natural fatty acids) to the mix.

The novelty of this study consists in exploring some of the qualitative aspects and mechanical proprieties of the PCL wood-based biopolymer parts generated by FDM with different printing conditions. The printing parameters varied in the experimental study were the printing temperature, the layer height, and the printing speed. The printing speed levels selected for the experiment were significantly superior to those usually tested in previous research in this field or those recommended by the filament producer.

The surface roughness of the parts was investigated. Higher Sa (arithmetical mean height) values were obtained when high printing temperatures and layer height were used. The surface texture obtained for these specific printing conditions exhibits signs of over-extrusion. Overall, the highest level chosen for the printing speed positively influenced the surface roughness.

Another novelty aspect of this study is the cutting machinability as a secondary machining operation of the FDM-printed wood-based composite biopolymer. Machinability is the property that characterizes the ease with which a material can be machined with a cutting tool. The machinability testing criteria used in this study was the cutting force components’ magnitude. The Biowood Rosa3D printed parts exhibit poor machinability in reference to natural wood parts (pinewood and beech wood). Therefore, lower cutting forces and better cutting machinability were obtained for the parts printed at higher printing temperatures and layer heights and with lower printing speeds.

Testing the capabilities of newly developed polymer composites, especially biopolymer composites, should be a constant concern for researchers to achieve competitive products for the industry. Many drawbacks of FDM 3D printing of wood-based biopolymers could be overcome by carefully choosing the processing parameters.

## Figures and Tables

**Figure 1 polymers-15-02305-f001:**
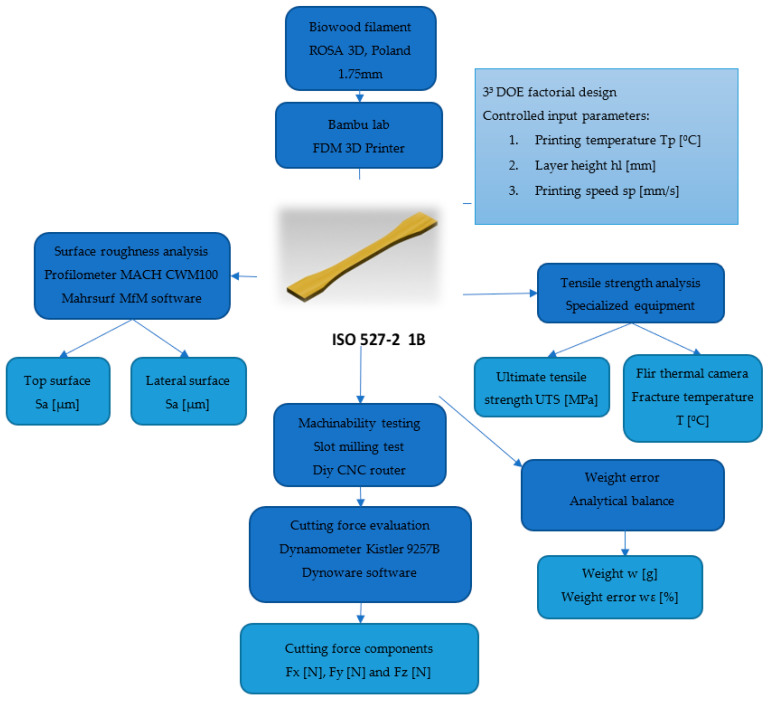
Schematic representation of the experimental program.

**Figure 2 polymers-15-02305-f002:**
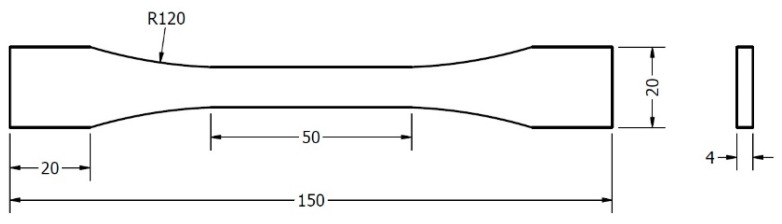
Tensile testing specimen ISO 527 Type 1B.

**Figure 3 polymers-15-02305-f003:**
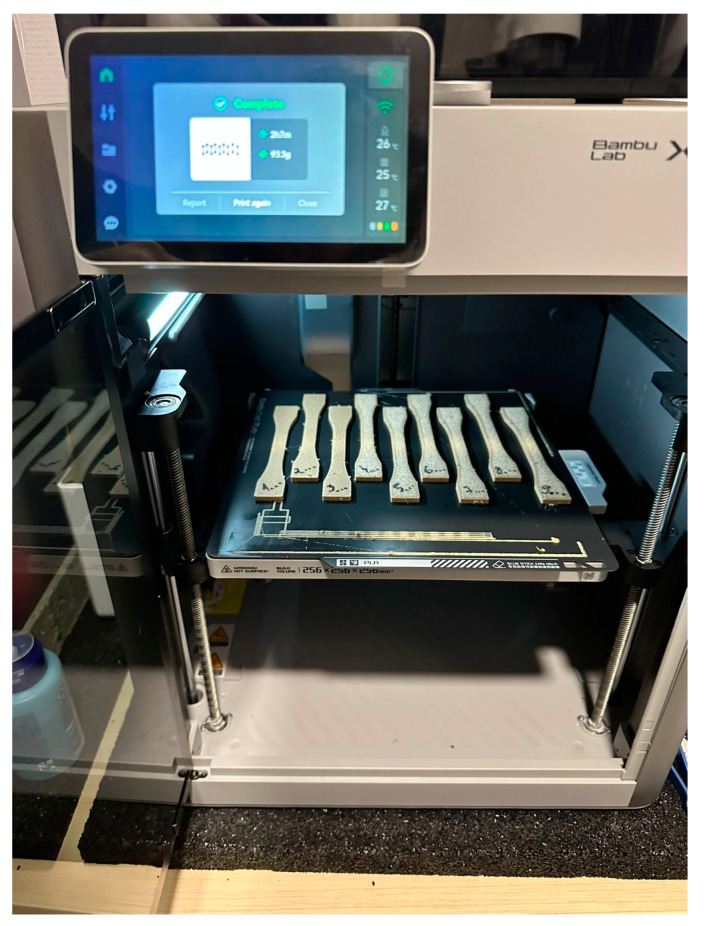
The Bambu Lab 3D FDM printer used in the experiments.

**Figure 4 polymers-15-02305-f004:**
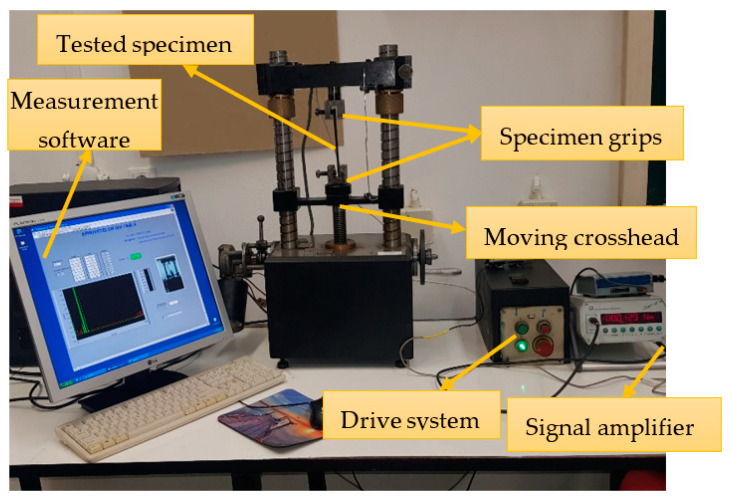
The ultimate tensile strength measuring device.

**Figure 5 polymers-15-02305-f005:**
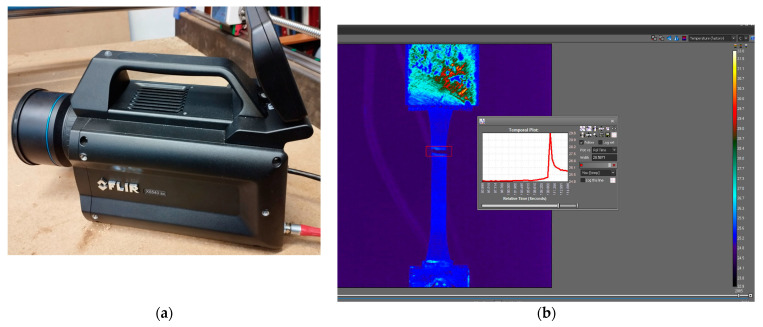
Fracture temperature measurement: (**a**) Flir X6540sc thermal camera; (**b**) fracture temperature analysis.

**Figure 6 polymers-15-02305-f006:**
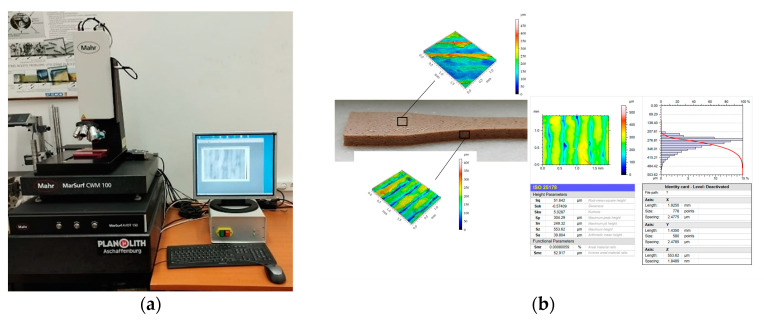
Surface roughness measuring procedure: (**a**) Mahr CWM 100 interferometer; (**b**) top and lateral surface analysis.

**Figure 7 polymers-15-02305-f007:**
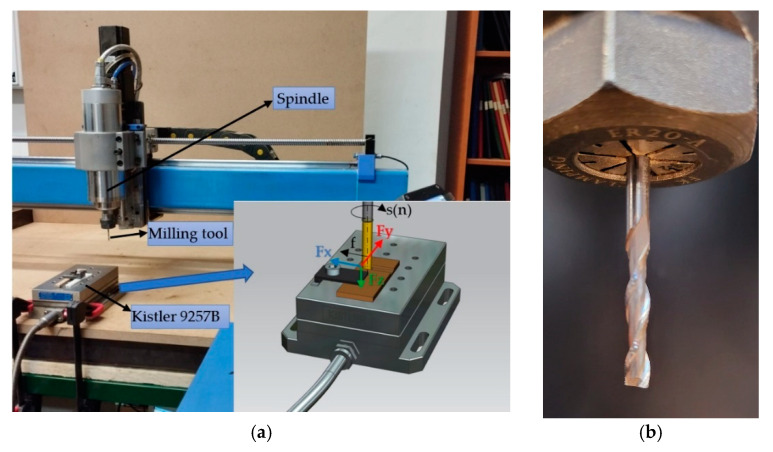
Slot milling tests setup: (**a**) cutting force measurement; (**b**) the end-milling tool.

**Figure 8 polymers-15-02305-f008:**
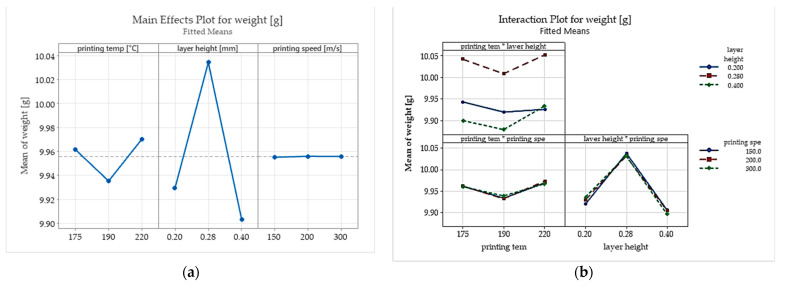
The influence of the selected input parameters on specimen weight: (**a**) the main effects plots; (**b**) interaction plots.

**Figure 9 polymers-15-02305-f009:**
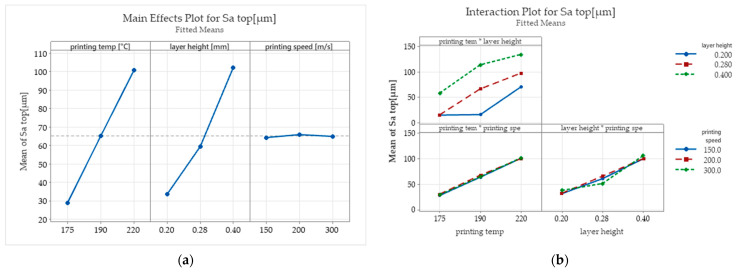
The influence exerted by the selected input parameters on Sa surface roughness of the top surface of the specimens: (**a**) the main effects plots; (**b**) interaction plots.

**Figure 10 polymers-15-02305-f010:**
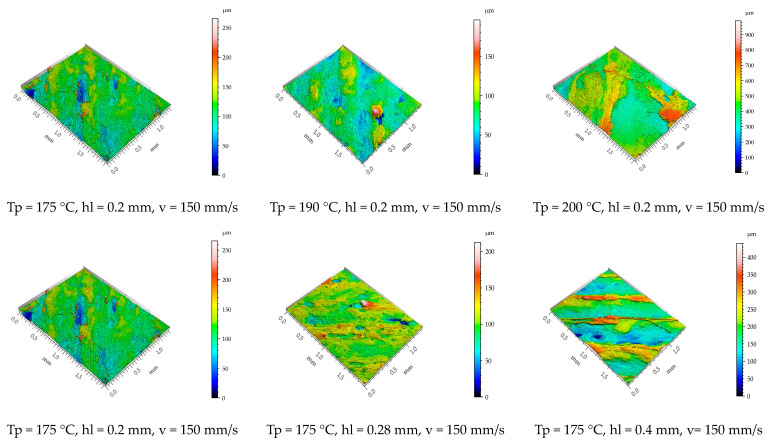
Isometric images top surface topography of the printed specimens.

**Figure 11 polymers-15-02305-f011:**
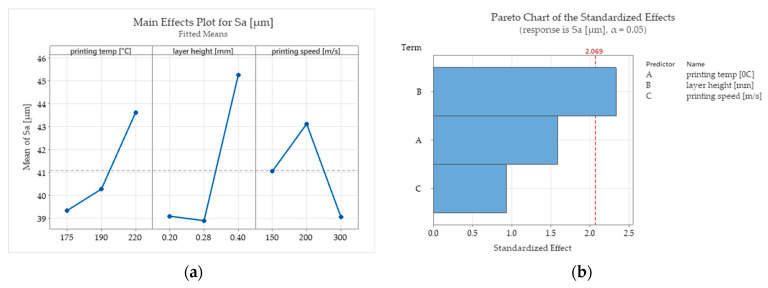
Main effects plots for Sa roughness parameter measured on the lateral surface of the specimens: (**a**) main effects plots for Sa; (**b**) Pareto chart.

**Figure 12 polymers-15-02305-f012:**
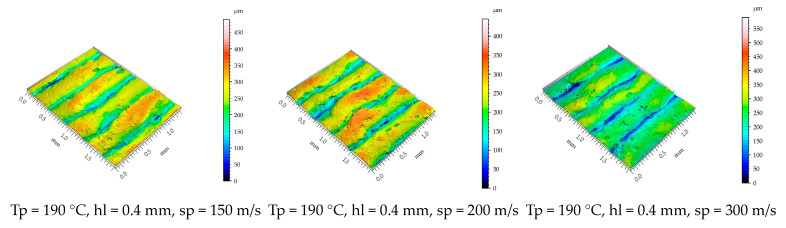
Isometric images of lateral surface topography of the printed specimens.

**Figure 13 polymers-15-02305-f013:**
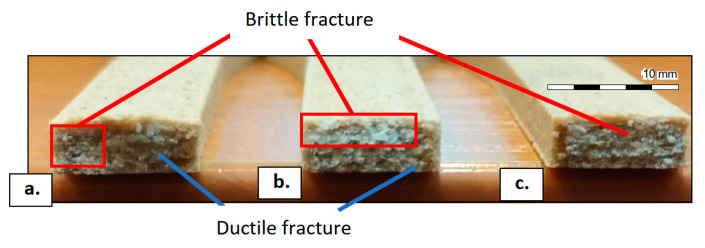
Fracture surfaces of the specimens after the tensile strength tests: (**a**) specimen printed at Tp = 175 °C with hl = 0.4 mm and sp = 200 mm/s; (**b**) specimen printed at Tp = 190 °C with hl = 0.4 mm and sp = 200 mm/s; (**c**) specimen printed at Tp = 220 °C with hl = 0.4 mm and sp = 200 mm/s.

**Figure 14 polymers-15-02305-f014:**
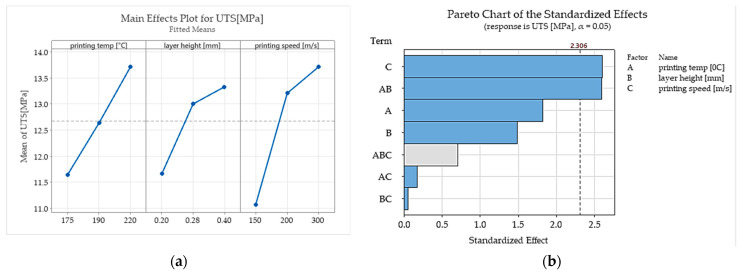
The influence of the selected input parameters over the ultimate tensile strength: (**a**) the main effects plots; (**b**) Pareto chart for the significance of the studied input parameters.

**Figure 15 polymers-15-02305-f015:**
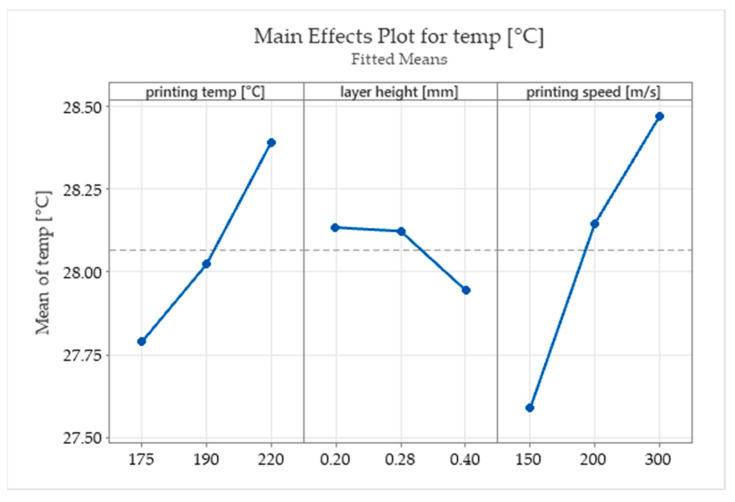
The main effects plots for the fracture temperature.

**Figure 16 polymers-15-02305-f016:**
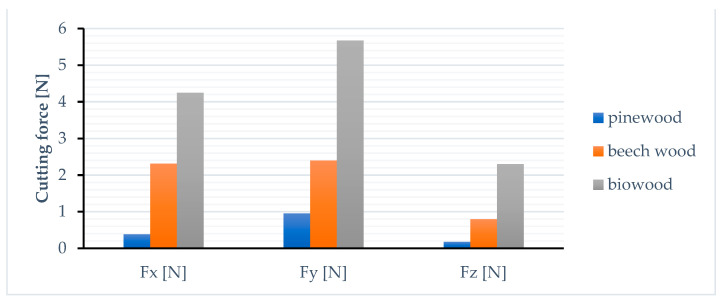
Cutting force components comparison between pinewood, beech wood, and Biowood parts generated during slot milling with cutting speeds of 150 m/min, cutting feeds of 800 mm/min, and cutting depth of 1.5 mm.

**Figure 17 polymers-15-02305-f017:**
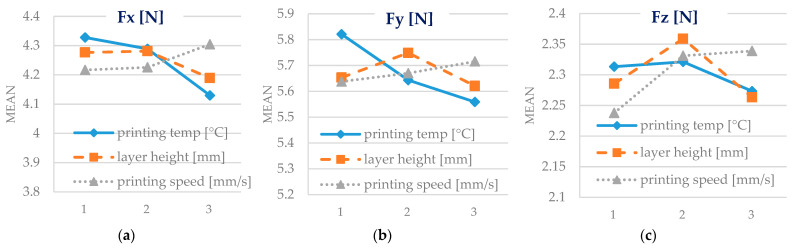
Mean effects plots for the cutting forces’ components. (**a**) main effect plot for Fx [N]; (**b**) main effect plot for Fy[N]; (**c**) main effect plot for Fz[N].

**Figure 18 polymers-15-02305-f018:**
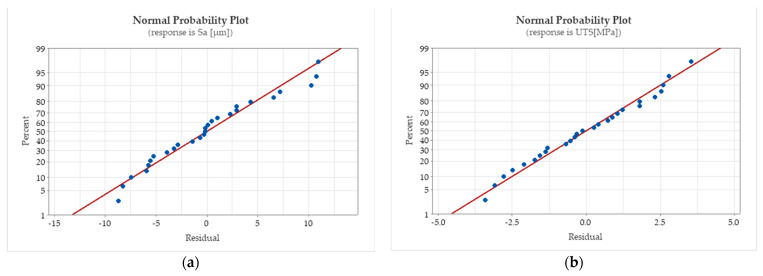
Normal probability plots for the nonlinear regressions (**a**) for the Sa roughness parameter measured on the lateral surfaces of the printed part and (**b**) for the ultimate tensile strength.

**Table 1 polymers-15-02305-t001:** Values of the input factors corresponding to the full factorial design.

Level	Input Parameters		
	Printing Temperature Tp (°C)	Layer Height hl (mm)	Printing Speed sp (mm/s)
1	175	0.2	150
2	190	0.28	200
3	220	0.4	300

**Table 2 polymers-15-02305-t002:** Physical properties of Biowood [34].

Softening point (°C)	50
Density (kg/m^3^)	1260
Elastic modulus (MPa)	3200
Tensile strength (MPa)	36

**Table 3 polymers-15-02305-t003:** Testing conditions and experimental results.

	Printing Temperature Tp (°C)	Layer Height hl (mm)	Printing Speed sp (mm/s)	Weight Error ε_w_ (%)	Top Surface Roughness, Sa (µm)	Lateral Surface Roughness Sal (μm)	Ultimate Tensile Strength, UTM (MPa)	Fracture Temperature T (°C)
1	175	0.2	150	7.100%	14.995	38.161	7.5000	28.1
2			200	6.971%	14.549	35.6655	13.1250	26.7
3			300	6.948%	14.441	35.343	13.2787	29.6
4		0.28	150	6.071%	13.845	40.418	10.0000	27.3
5			200	6.108%	16.238	33.848	11.4062	27.5
6			300	6.056%	14.034	37.807	12.4992	28.1
7		0.4	150	7.328%	55.260	46.174	12.5000	27.2
8			200	7.403%	59.768	48.912	12.7840	28.7
9			300	7.509%	58.225	37.840	11.7187	26.9
10	190	0.2	150	7.262%	14.9585	45.719	9.2160	26.7
11			200	7.286%	16.599	36.305	12.3437	29.2
12			300	7.130%	15.560	36.400	15.3125	27.1
13		0.28	150	6.379%	62.972	44.391	13.1250	28.2
14			200	6.413%	67.723	35.031	15.7824	27.9
15			300	6.388%	67.476	31.971	15.4688	28.9
16		0.4	150	7.619%	113.845	36.211	9.0624	27.3
17			200	7.608%	118.236	43.638	11.5625	27.6
18			300	7.605%	109.852	52.905	11.8750	29.3
19	220	0.2	150	7.260%	64.635	38.328	12.0313	28
20			200	7.117%	64.556	42.606	11.4062	28.4
21			300	7.119%	82.446	43.400	10.7812	29.4
22		0.28	150	5.906%	106.59	36.131	12.0313	27.7
23			200	6.004%	112.97	54.213	13.2800	28.5
24			300	6.049%	72.425	36.364	13.4375	29
25		0.4	300	7.098%	131.640	44.051	14.2188	27.8
26			150	7.045%	122.390	57.922	17.1872	28.8
27			200	7.164%	149.720	39.632	19.0624	27.9

**Table 4 polymers-15-02305-t004:** Analysis of Variance for weight best fit regression.

Source	DF	Adj SS	Adj MS	F-Value	*p*-Value
Regression	3	0.008754	0.002918	0.77	0.521
printing temp (°C)	1	0.001036	0.001036	0.27	0.605
layer height (mm)	1	0.007717	0.007717	2.04	0.166
printing speed (m/s)	1	0.000001	0.000001	0.00	0.986
Error	23	0.086840	0.003776		
Total	26	0.095594			

**Table 5 polymers-15-02305-t005:** Analysis of Variance for Sa top best fit regression.

Source	DF	Adj SS	Adj MS	F-Value	*p*-Value
Regression	3	43,847.9	14,616.0	62.55	0.000
printing temp (°C)	1	22,324.0	22,324.0	95.53	0.000
layer height (mm)	1	21,523.4	21,523.4	92.11	0.000
printing speed (m/s)	1	0.5	0.5	0.00	0.964
Error	23	5374.7	233.7		
Total	26	49,222.6			

**Table 6 polymers-15-02305-t006:** Analysis of Variance for Sa.

Source	DF	Adj SS	Adj MS	F-Value	*p*-Value
Regression	3	313.20	104.40	2.95	0.054
printing temp (°C)	1	89.57	89.57	2.53	0.125
layer height (mm)	1	192.69	192.69	5.44	0.029
printing speed (m/s)	1	30.95	30.95	0.87	0.360
Error	23	814.71	35.42		
Total	26	1127.91			

**Table 7 polymers-15-02305-t007:** Analysis of Variance for UTS (MPa).

Source	DF	Adj SS	Adj MS	F-Value	*p*-Value
Regression	3	56.29	18.763	4.19	0.017
printing temp (°C)	1	18.75	18.748	4.19	0.052
layer height (mm)	1	11.32	11.321	2.53	0.125
printing speed (m/s)	1	26.22	26.220	5.86	0.024
Error	23	102.95	4.476		
Total	26	159.24			

**Table 8 polymers-15-02305-t008:** The parameter estimation errors for Equation (2).

Parameter	Estimate	SE Estimate
×1	7.31547	12.0722
×2	0.45580	0.2951
×3	0.22107	0.1002
x4	−0.07472	0.0998

**Table 9 polymers-15-02305-t009:** The parameter estimation errors for Equation (3).

Parameter	Estimate	SE Estimate
×1	0.088580	0.161801
×2	0.713767	0.326272
×3	0.197467	0.111027
x4	0.270825	0.109504

**Table 10 polymers-15-02305-t010:** The parameter estimation errors for Equation (4).

Parameter	Estimate	SE Estimate
×1	9.58440	0.686965
×2	0.00615	0.012852
×3	−0.00408	0.004299
x4	0.00008	0.004279

## Data Availability

Some or all data, models, or code generated or used during the study are available from the corresponding author by request.

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
