# Peer review of "Influence of 3D Printing Conditions on Some Physical–Mechanical and Technological Properties of PCL Wood-Based Polymer Parts Manufactured by FDM"

_polymers, 2023, doi:10.3390/polym15102305_

Round 1

Reviewer 1 Report

This paper analysis the tensile strength, surface quality, and cutting machinability as secondary processing operation of PCL wood-based polymer parts generated by FDM 3D printing. Also, using the design of the experiment, the effect of printing parameters has been checked. The article's title is practical and attractive, but the following points should be considered before publishing.

What is the justification for using the machining process for 3D-printed parts? As you know, one of the main advantages of 3D printing is the production of integrated and complex parts. Meanwhile, with the FDM process, it is possible to produce almost complex parts. The purpose of conducting this research and examining the machining of 3D-printed parts with biocompatibility needs justification.

The abstract should be written better and needs major revisions. The purpose of research and innovation should be clearly stated. Also, the performed tests should be presented first, and then the results should be presented quantitatively and qualitatively.

The article needs general writing and grammar editing. References should also be numbered in order (Lines 70, 99, etc.).

The introduction is written very briefly, and at the end, a suitable summary of the importance of the present issue is not provided. Use the following resources to complete this section. 3D printing of PLA-TPU with different component ratios: Fracture toughness, mechanical properties, and morphology.  Development of Pure Poly Vinyl Chloride (PVC) with Excellent 3D Printability and Macroand MicroStructural Properties.

How to choose printing parameters should also be mentioned. It seems that the selected parameters are not very suitable. Refer to the following source and use it for comparison. 4D PrintingEncapsulated Polycaprolactone–Thermoplastic Polyurethane with High Shape Memory Performances. PCL has a melting temperature of about 60 degrees Celsius, and printing is usually done below 150 degrees Celsius. On the other hand, the presented printing speeds are not logical, and the unit seems wrongly mentioned.

Also, why are the selected parameters not regular? Usually, it is essential to consider the middle point of the selected interval for DOE if this issue is not seen in all three parameters.

The results section is well organized and categorized. But some parts report the results, which require corrections and deepening the analysis and discussion.

The scale bar and error bar is missing for some images and results.  Use the recommended sources to analyze the results. It is suggested to modify the conclusion section as well as the abstract.

No comment.

Author Response

Reviewer's comment no. 1. What is the justification for using the machining process for 3D-printed parts? As you know, one of the main advantages of 3D printing is the production of integrated and complex parts. Meanwhile, with the FDM process, it is possible to produce almost complex parts. The purpose of conducting this research and examining the machining of 3D-printed parts with biocompatibility needs justification.

Author’s response to the reviewer's comment: A detailed justification was included in the revised paper, lines 152-162.

Reviewer's comment no. 2. The abstract should be written better and needs major revisions. The purpose of research and innovation should be clearly stated. Also, the performed tests should be presented first, and then the results should be presented quantitatively and qualitatively.

Author’s response to the reviewer's comment: The abstract was reformulated, and, in the Introduction section, additional explanations regarding the content of the article has been added. See revised paper, lines 12-25 and 137-151.

Reviewer's comment no. 3. The article needs general writing and grammar editing. References should also be numbered in order (Lines 70, 99, etc.).

Author’s response to the reviewer's comment: The paper underwent English proofing, and the reference order was corrected. See revised paper.

Reviewer's comment no. 4. The introduction is written very briefly, and at the end, a suitable summary of the importance of the present issue is not provided. Use the following resources to complete this section. 3D printing of PLA-TPU with different component ratios: Fracture toughness, mechanical properties, and morphology. A New Strategy for Achieving Shape Memory Effects in 4D Printed Two-Layer Composite Structures. Development of Pure Poly Vinyl Chloride (PVC) with Excellent 3D Printability and Macro‐and Micro‐Structural Properties.

Author’s response to the reviewer's comment. The suggestion has been taken into consideration. The Introduction section was developed and other references to the study were introduced, among those, some indicated by the reviewer.

Reviewer's comment no. 5. How to choose printing parameters should also be mentioned. It seems that the selected parameters are not very suitable. Refer to the following source and use it for comparison. 4D Printing‐Encapsulated Polycaprolactone–Thermoplastic Polyurethane with High Shape Memory Performances. PCL has a melting temperature of about 60 degrees Celsius, and printing is usually done below 150 degrees Celsius. On the other hand, the presented printing speeds are not logical, and the unit seems wrongly mentioned.

Author’s response to the reviewer's comment: The suggestion was taken into consideration, and the specified source was used. Biowood Rosa 3D filament producers recommend using printing temperatures of 170-210 °C, printing speeds in the range 60-80mm/s and printing bed temperatures of 30-50 °C, see revised paper, lines 235-237 and 210-225. The printing speeds were chosen according to the 3D printer used, which is a BambuLab X1C, a Core XY printer, capable to print at 500 mm/s. The units for printing speed are millimetres per second. The terminology used (ps) was chosen in this case because the current standard (ISO/ASTM 52900- Additive manufacturing — General principles — Fundamentals and vocabulary) does not offer a standardised terminology for this parameter.

Reviewer's comment no. 6. Also, why are the selected parameters not regular? Usually, it is essential to consider the middle point of the selected interval for DOE if this issue is not seen in all three parameters.

Author’s response to the reviewer's comment: A complete explanation was introduced in the revised paper, lines 210-225.

Reviewer's comment no. 7: The results section is well organized and categorized. But some parts report the results, which require corrections and deepening the analysis and discussion.

Author’s response to the reviewer's comment: The revised paper introduced tables 4, 5, 6 and 7 with ANOVA analysis and paragraphs with conclusions drawn from these analyses. Furthermore,  the standard error of the regression S, the correlation coefficient R and the parameters estimation errors for each regresion were introduced. Also, for the regression equations 2 and 3, Normal probability plots were introduced and discussed. The revised paper introduced section 4 Discutions and new paragraphs. which includes a critical analysis based on the results.

Reviewer's comment no. 8: The scale bar and error bar is missing for some images and results.  Use the recommended sources to analyze the results. It is suggested to modify the conclusion section as well as the abstract.

Author’s response to the reviewer's comment: The same scale was used in Figures 10 and 12 and the axis titles for the graph from figure 16 was introduced. Also, end milling conditions were stipulated in the figure title. Section 5, Conclusions, and abstract was reconsidered.

Reviewer 2 Report

Notes in the attachment.

Author Response

Reviewer's comment no. 1: The subject of the article does not correspond to the scope of the research. Please change the subject of the article.

Author’s response to the reviewer's comment: The authors consider that the reviewer was right. The following title for the paper “Influence of 3D printing conditions on some physical-mechanical and technological properties of PCL wood-based polymer parts manufactured by FDM” is proposed.

Reviewer's comment no. 2. Please explain exactly the purpose of the research. The impact of 3D printing output parameters on what was analyzed? Rm, Sa, machinability aspects?

Author’s response to the reviewer's comment: The introduction section was elaborated, giving more clarity regarding the purpose of the research. See revised paper, lines 138-152.

Reviewer's comment no. 3. Why was the Sa parameter used to evaluate the surface topography? Please explain.

Author’s response to the reviewer's comment: The authors considered that the reviewer's question is justified. Explanations were introduced regarding the arguments that led to the consideration of the Sa parameter for the characterization of the surface topography of the 3D printed parts. See revised paper, lines 285-296.

Reviewer's comment no. 4. In Figure 2, this is not a pipe.

Author’s response to the reviewer's comment: The reviewer was right. The term "Tensile testing tube" has been replaced by "Tensile test specimen".

Reviewer's comment no. 5. In Figures 10 and 12, please use the same scale.

Authors response to the reviewer's comment: Figures 10 and 12 were revised and the same scale was used.

Reviewer's comment no. 6. Figure 16. Please describe the axes. The drawing is illegible.

Author’s response to the reviewer's comment: Figure 16 was revised, and the axis titles for the graph was introduced. Also, end milling conditions were stipulated in the figure title.

Reviewer's comment no. 7. What cutting parameters were used in the tests?

Author’s response to the reviewer's comment: More clarity on the cutting parameters and cutting tool was introduced into the revised paper. See revised paper, lines 310-316.

Reviewer's comment no. 8. Please describe the geometry of the cutting tool and the material of the blade.

Authors response to the reviewer's comment: The reviewer was right. More clarity on the cutting tool geometry and material was introduced into the revised paper. See lines 310-316 and figure 7.b.

Reviewer's comment no. 9. Please write what are the parameters estimation errors according to the equations: 2, 3, 4.

Authors response to the reviewer's comment: In the revised paper, tables 8, 9 and 10 were introduced, that presents the parameters estimation errors.

Reviewer's comment no. 10. I propose to perform a simple ANOVA analysis for the tested parameters.

Authors response to the reviewer's comment: The revised paper introduced ANOVA analysis for the obtained results (table 4, 5, 6 and 7) and also paragraphs with conclusions drawn from these analyses.

Reviewer's comment no. 11. Equation No. 1 is not marked.

Authors response to the reviewer's comment: The reviewer was right. The equation marking was corrected.

Reviewer 3 Report

The paper needs the following revisions before publication:

1.     The novelty/objective of the paper should be highlighted in more detail. Hence, more focus should be given to the last paragraph of the introduction.

2.     Although the authors have discussed many studies in the literature. However, due to the increasing interest in different industries, the scientific community devotes remarkable efforts to these applications. Therefore, referring to and including the recent paper in the literature is worth mentioning: https://doi.org/10.3390/app13020904.

3.     Please check the Caption of the Tables.

4.     Experimental results must not be part of the experimental section.

5.     The details of Figure 1 in the text are not available. Please check the sequence of all the Tables/Figures and assign them in appropriate places with relevant captions and numbering.

6.     I recommend using the word “tensile specimen” instead of the dog-bone specimen.

7.     The specific name of the tensile strength measuring device should be included in the paper.

8.     Lines 226 – 232 should be part of the Materials and methods/experimental setup.

9.     The experimental results should be discussed with separate subheadings in section 3, such as tensile testing, machinability testing, surface roughness analysis etc.

10. The errors in the mathematical equations should be mentioned in the paper. It should also be more convenient for the readers to present the experimental results and the values obtained from the equations in graphs.

11. A separate discussion section is required, which includes a critical analysis based on the results.

12. The conclusions are too lengthy and should be focused more on the results of this study with future recommendations and industrial applications.

13. Extensive English proofreading is required. The use of personal pronouns should be avoided in technical papers.

Extensive editing of the English language is required.

Author Response

Reviewer's comment no. 1. The novelty/objective of the paper should be highlighted in more detail. Hence, more focus should be given to the last paragraph of the introduction.

Author’s response to the reviewer's comment. The introduction section was further elaborated and a clearer insight on the novelty/objective of the paper was introduced. See revised paper, lines 138-167.

Reviewer's comment no. 2. Although the authors have discussed many studies in the literature. However, due to the increasing interest in different industries, the scientific community devotes remarkable efforts to these applications. Therefore, referring to and including the recent paper in the literature is worth mentioning: https://doi.org/10.3390/app13020904.

Author’s response to the reviewer's comment. In the revised paper, the Introduction section was further developed, with new bibliographic references, including the one indicated by the reviewer.

Reviewer's comment no. 3. Please check the Caption of the Tables.

Author’s response to the reviewer's comment. The table captions were corrected.

Reviewer's comment no. 4. Experimental results must not be part of the experimental section.

Author’s response to the reviewer's comment. The authors felt that the reviewer was right. Table 2 with the experimental results was moved to section 3 Results.

Reviewer's comment no. 5. The details of Figure 1 in the text are not available. Please check the sequence of all the Tables/Figures and assign them in appropriate places with relevant captions and numbering.

Author’s response to the reviewer's comment. The sequence of all the tables/figures was corrected. A new paragraph describing the research steps presented in figure 1 was inserted. See revised paper, lines 169-185.

Reviewer's comment no. 6. I recommend using the word “tensile specimen” instead of the dog-bone specimen.

Author’s response to the reviewer's comment. The reviewer was right. The term "dog-bone specimen" has been replaced by "Tensile test specimen".

Reviewer's comment no. 7. The specific name of the tensile strength measuring device should be included in the paper.

Author’s response to the reviewer's comment: The tensile strength of the specimens was measured using an experimental set-up (Figure 4), previously built within the Faculty of Mechanical Engineering, Automotive and Robotics from the “Stefan cel Mare” University of Suceava, Romania. Therefore, the equipment has no specific name.

Reviewer's comment no. 8. Lines 226 – 232 should be part of the Materials and methods/experimental setup.

Author’s response to the reviewer's comment. The mentioned paragraph was moved to section 2.2, Sample preparation and equipment.

Reviewer's comment no. 9. The experimental results should be discussed with separate subheadings in section 3, such as tensile testing, machinability testing, surface roughness analysis etc.

Author’s response to the reviewer's comment. In the revised paper, section 3 was divided into subsections as reviewer indicated.

Reviewer's comment no. 10. The errors in the mathematical equations should be mentioned in the paper. It should also be more convenient for the readers to present the experimental results and the values obtained from the equations in graphs.

Author’s response to the reviewer's comment. The authors felt that the reviewer was right. We had intruduced the standard error of the regression S, the correlation coefficient R and the parameters estimation errors for each regresion. Also, for the regressions from equations 2 and 3 Normal probability plots were introduced.

Reviewer's comment no. 11. A separate discussion section is required, which includes a critical analysis based on the results.

Author’s response to the reviewer's comment. The revised paper introduces section 4, Discutions, which includes new paragraphs. See revised paper.

Reviewer's comment no. 12. The conclusions are too lengthy and should be focused more on the results of this study with future recommendations and industrial applications.

Author’s response to the reviewer's comment. Section 5, Conclusions, was reconsidered.

Reviewer's comment no. 13. Extensive English proofreading is required. The use of personal pronouns should be avoided in technical papers.

Author’s response to the reviewer's comment. The revised paper underwent extensive English proofreading, including the used Grammarly premium software for English editing.

Round 2

Reviewer 2 Report

The authors answer my questions correctly. The article has been corrected. Thank you.

Reviewer 3 Report

The authors have significantly revised the paper, and I recommend the paper for publication. Thanks.

Minor editing of English is required